# Benchmarking PASADENA Consensus along the Learning Curve of Robotic Radical Cystectomy with Intracorporeal Neobladder: CUSUM Based Assessment

**DOI:** 10.3390/jcm10245969

**Published:** 2021-12-19

**Authors:** Riccardo Lombardo, Riccardo Mastroianni, Gabriele Tuderti, Mariaconsiglia Ferriero, Aldo Brassetti, Umberto Anceschi, Salvatore Guaglianone, Cosimo De Nunzio, Antonio Cicione, Andrea Tubaro, Michele Gallucci, Giuseppe Simone

**Affiliations:** 1Ospedale Sant’Andrea, ‘Sapienza’ Universita di Roma, 00189 Rome, Italy; acicione@libero.it; 2Department of Urology, IRCCS “Regina Elena” National Cancer Institute, 00121 Rome, Italy; rmastroianni@gmail.com (R.M.); gtuderti@gmail.com (G.T.); ferriero@gmail.com (M.F.); brassettialdo@gmail.com (A.B.); anceschiu@gmail.com (U.A.); guaglianone@gmail.com (S.G.); cosimodenunzio@virgilio.it (C.D.N.); andreatubaro@gmail.com (A.T.); gallucci@gmail.com (M.G.); puldet@gmail.com (G.S.)

**Keywords:** bladder cancer, urinary diversion, ileal conduit, CUSUM, neobladder

## Abstract

(1) Aim: Robot assisted radical cystectomy (RARC) with intacorporeal neobladder (iN) is a challenging procedure. There is a paucity of reports on RARC-iN, the extracorporeal approach being the most used. The aim of our study was to assess the learning curve of RARC-iN and to test its performance in benchmarking Pasadena consensus outcomes. (2) Material and methods: The single-institution learning curve of RARC-iN was retrospectively evaluated. Demographic, clinical and pathologic data of all patients were recorded. Indications to radical cystectomy included muscle invasive bladder cancer (pT ≥ 2) or recurrent high grade non muscle invasive bladder cancer. The cumulative sum (CUSUM) technique, one of the methods developed to monitor the performance and quality of the industrial sector, was adopted by the medical field in the 1970s to analyze learning curves for surgical procedures. The learning curve was evaluated using the following criteria: 1. operative time (OT) <5 h; 2. 24-h Hemoglobin (Hb) drop <2 g/dl; 3. severe complications (according to the Clavien classification system) <30%; 4. positive surgical margins <5%; and 5. complete lymph-node dissection defined as more than 16 nodes. Benchmarking of all five items on quintile analysis was tested, and a failure rate <20% for any outcome was set as threshold. (3) Results: the first 100 consecutive RARC-iN patients were included in the analysis. At CUSUM analysis, RARC required 20 cases to achieve a plateau in terms of operative time (defined as more than 3 consecutive procedures below 300 min). Hemoglobin drop, PSM and number of removed lymph-nodes did not change significantly along the learning curve. Overall, 41% of the patients presented at least one complication. Low-grade and high-grade complication rates were 30% and 17%, respectively. When assessing the benchmarks of all five Pasadena consensus outcomes on quintile analysis, a plateau was achieved after the first 60 cases. (4) Conclusions: RARC-iN is a challenging procedure. The potential impact of the learning curve on significant outcomes, such as high grade complications and positive surgical margins, has played a detrimental effect on its widespread adoption. According to this study, in tertiary referral centers, 60 procedures are sufficient to benchmark all outcomes defined in Pasadena RARC consensus.

## 1. Introduction 

In 2021, 83,730 cases of bladder cancer are estimated to have occurred worldwide, with 17,200 deaths [1]. Bladder cancer represents the sixth neoplasm () ion the world. The overall five year survival rate is 77.1%; however, it drops to 37% when regional lymph-nodes are involved. Patients with localized disease represent 75%, and in younger patients the prevalence is even higher. The long-term survival of patients with pT1 and pTis disease explains the high prevalence and lower risk of cancer specific survival when compared to T2-T4 tumours [2]. 

Radical cystectomy with lymph-node dissection +/− neoadjuvant chemotherapy represents the gold standard for the treatment of recurrent BCG resistant pT1 disease and T2-T4 tumours. In young patients with good performance status, orthotopic neobladder is often the urinary diversion of choice. However, early and late comorbidity is as high as 22%, and long-term complications include diurnal (8–10%) and nocturnal (20–30%) incontinence, uretero-intestinal stenosis (3–18%), metabolic disorders and B12 deficiency [3,4,5,6]. 

The introduction of robotic surgery aimed to improve surgical outcomes of RC with orthotopic neobladder in recent years [7,8,9]. Although the data is still insufficient, RCT comparing open and robotic surgery suggests that robotic surgery is associated with shorter hospital stays (median 1 d), lower blood loss and longer operative times when compared to open surgery [10,11,12]. Robotic platforms are increasing all over the world and therefore some authors have evaluated the learning curve of RRC with orthotopic neobladder with conflicting results. More specifically, according to the literature, 25–250 cases are needed to reach proficiency; however, the Pasadena Consensus panel suggested a benchmark of 30 cases. 

The assessment of the learning curve of a surgical procedure is of outmost importance. Although some studies have evaluated the learning curve of RRC with orthotopic neobladder, these studies assess few outcomes and lack standardized methods of learning curve assessment. In patients undergoing RC with orthotopic neobladder it is important to evaluate operative time, blood loss, severe complications, surgical margins and number of lymph-nodes retrieved as suggested by the Pasadena consensus panel. A standardized method to assess the learning curve of surgical procedures is the CUSUM method, which has been validated in several surgeries [13,14,15,16,17,18,19]. 

With this knowledge in mind, the aim of our study was to assess learning curve of RRC with intracorporeal orthotopic neobladder using the CUSUM method. 

## 2. Materials and Methods 

A series of patients undergoing robotic radical cystectomy with intracorporeal neobladder were consecutively enrolled. All patients signed an informed consent, and the study was carried out in accordance with the Helsinki declaration. Indications for surgery were muscle invasive bladder cancer or recurrent non-muscle invasive bladder cancer resistant to BCG. Exclusion criteria included extra-vesical extension and bladder cancer in the prostatic urethra and/or bladder neck. 

Demographic characteristics including the perioperative, operative and postoperative data of the patients were collected. More specifically, age, sex, height, weight, smoking status, hypertension history, diabetes history, chronic kidney disease and ASA score were recorded. As well, operative time and intraoperative complications were recorded. Lastly, time to flatus, time to bowel, time to oral intake and postoperative complications were recorded. 

Complications were recorded and classified according to the modified Clavien-Dindo classification and divided into peri-operative, intra-operative, early post-operative (within 30 days) and late-pos-operative complications. 

### 2.1. Surgical Technique 

All RARC were performed by a single expert open, laparoscopic and robotic surgeon over 65 years old with an extensive previous background including more than 1000 robotic procedures and more than 1000 open cystectomies. The technique was not modified along the learning curve; ICG and low-pressure pneumoperitoneum was not available. The interventions were performed under general anesthesia and the fast-track ERAS protocol was applied as described by Karl et al. [20]. Port placement is shown in Figure 1. Besides the standard ports an additional port was placed for J-J stents placement. RARC was performed according to the technique described by Desai et al. For oncological purposes, distal ureters were sent for frozen section analysis. In order to avoid urine spillage a Hem-o-lock is placed on the urethra before its incision. Extended lymph-node dissection was performed and sent for analysis in different packages according to their anatomical location (obturator, internal, external, common iliac and presacral nodes). 

The division of the proximal ileum was made using only one stapler load (60 mm). Isolation of the distal extremity of the ileal segment was carried out with a 6- to 8-cm deep section of the mesentery using two consecutive stapler loads (60 mm and 45 mm).

The neobladder was configurated using 42 cm of ileum, the ileum segment was chosen at least 20 cm away from the ileocecal valve (Figure 2). A tension-free approach was used to create the neo-bladder neck. The dome and the base of the neobladder is configured with the proximal half of the loop.

The proximal half of the loop was used to configure the left base and the dome of the neobladder.

The bladder neck of the neobladder is created in a U-shaped manner leaving 8 cm and 18 cm on each side. As shown in Figure 2, the ileum is de-tubularized and then the neobladder is configured using motorized staplers. The neobladder neck is then anastomized to the urethra using a two end-knotted 2-0 Monocryl Visi-Black running suture. Finally, a 22 F catheter is introduced and inflated with 5 mL of saline solution. The anastomosis of the spatulated ureters is performed posteriorly with a 4-0 Monocryl (Ethicon, Somerville, NJ, USA) interrupted suture. JJ stents are inserted through the pre-pubic trocar.

Intestinal anastomosis was performed.

A single uro-pathologist analyzed the specimen and staging was made according to the 2009 American Joint Committee on cancer codification system [21].

### 2.2. Learning Curve Assessment

The cumulative sum (CUSUM) technique, one of the methods developed to monitor the performance and quality of the industrial sector, was adopted by the medical field in the 1970s to analyze learning curves for surgical procedures. The learning curve was evaluated using the following criteria: Operative time (OT) <5 h;24-h Hemoglobin (Hb) drop <2 g/dl;Severe complications (according to the Clavien classification system) <30%;Positive surgical margins <5%;Complete lymph-node dissection (defined as more than 16 nodes).

Benchmarking of all five items on quintile analysis was tested, and a failure rate <20% for any outcome was set as threshold. 

In order to detect a shift in the trend of OT and HS, the CUSUM technique was used. Cases were arranged in chronological order. CUSUMOT 1st is the difference between OT1st and the mean OT, i.e., OT1st−OTmean. CUSUMOT 2nd is calculated as CUSUMOT 1st + (OT2nd−OTmean). CUSUMOT 3rdis calculated as CUSUMOT 2nd + (OT3rd−OTmean). The following equation illustrates this recursive process.
CUSUMOT*x* = CUSUMOT*x* − 1 + (OT*x* − OTmean)

#### Graph Interpretation

The CUSUM value is plotted on the y-axis against the number of procedures on the x-axis. The CUSUM plotted line is a running sum of increments (1- S) and decrements (S). Therefore, if the plotted line crosses the control line in an upward trend, performance is deemed unacceptable. If the control line is crossed in a downward trend, then performance is deemed acceptable. If performance is maintained between two control lines, then acceptable performance is being maintained. Competence is declared when two consecutive control lines are crossed in a downward fashion. 

### 2.3. Statistical Analysis

Statistical analysis was performed using the Statistical Package for the Social Sciences (SPSS v.24, IBM Corp., Armonk, NY, USA). Evaluation of data distribution using the Kolmogorov–Smirnov test showed a non-normal distribution of the study data set. Differences between surgeons in medians for quantitative variables and differences in distributions for categorical variables were tested with the Kruskal Wallis one-way analysis of variance and chi-square test, respectively. The learning curve was assessed with the CUSUM analysis for the continuous variables. The population was divided in quintiles and a cut-off of 20% was applied as a benchmark for each variable. Operative time was considered adequate if <300 min, Hb drop if <2 g/dl and lymph node count >16 LND. 

## 3. Results 

The first 100 consecutive RARC-iN patients were included in the analysis. The general characteristics of the enrolled population are described in Table 1. All the procedures were completed successfully with a median operative time of 330 min (300/378) and none of the patients was converted to open surgery.

### 3.1. Operative Time

Overall median operative time along the first 100 interventions was 330 min (300/378). At CUSUM analysis, RARC required 20 cases to achieve a plateau in terms of operative time (defined as more than three consecutive procedures below 300 min). When looking at the first 10 procedures performed, operative time is highly variable, reaching a maximum operative time of 640 min (Figure 1). 

### 3.2. Complications 

Overall, 26 patients presented a postoperative complication; however, only 12 of them presented a high-grade complication (Clavien >II). Complications are listed in Table 2. Median length of stay was 10 (9/14), however 33/100 patients required readmission. Reasons for readmission are listed in Table 3. When looking at the learning curve in terms of quintiles, it takes at least 40 procedures to obtain a Clavien >II complication rate bellow 20%. In the first 20 cases the complication rate is as high as 35% (Figure 3). 

### 3.3. Haemoglobin Drop

Overall median Hemoglobin drop among the first 100 cases was 2.7 (2.0/3.7). Although no statistical difference was recorded between quintiles, in the first 40 cases the rate of Hb drop >2 g/dl was >30% while in the last 60 cases <30% (Figure 2). At CUSUM analysis, the Hb drop did not change significantly along the learning curve. 

### 3.4. Positive Surgical Margins

Overall, 10/100 patients presented positive surgical margins. All positive surgical margins were found on the ureters which were sent for frozen section while no soft tissue positive surgical margins were recorded. No statistically significant differences were recorded in terms of positive surgical margins between quintiles. However, in the last 20 cases none of the patients presented positive surgical margins. 

### 3.5. Lymphnode Count 

The overall median number of removed lymph-nodes was 32 (24/38). The number of LND did not change significantly along the learning curve. However, in the last 20 cases all the patients presented a number of LND > 16 (Figure 2). 

### 3.6. Comprehensive Analysis

When assessing benchmark of all five Pasadena consensus outcomes on quintile analysis, a plateau was achieved after the first 60 cases. More specifically, in the first and the second quintiles, no more than two criteria are satisfied. The third quintile satisfies four out of five criteria, while the last two satisfy all five criteria (Figure 3). 

## 4. Discussion 

In the present study evaluating the learning curve of robotic radical cystectomy, we established a cut-off of 60 procedures to satisfy the five criteria established by the Pasadena consensus panel. When looking at our data it is important to consider that the learning curve was assessed in an expert open and laparoscopic surgeon. According to our results, Hb drop, PSM and LND count do not change significantly along the learning curve, however, high-grade complications and operative time are high particularly in the first 40 cases. Surgeons approaching robotic RC should keep in mind these results before starting robotic surgery. 

The learning curve of RC with intracorporeal urinary diversion has been explored by some authors, with conflicting results. In 2010, Hayn et al. evaluated the learning curve of RARC using the international robotic cystectomy consortium database, demonstrating an acceptable level of proficiency after 30 cases, however their analysis did not consider the type of urinary diversion [22]. In 2019 Porreca et al. evaluated the learning curve of the first 100 procedures of RARC with urinary diversion. The authors divided the cases into three groups to assess the learning curve, however they found no statistically significant differences between groups in terms of blood loss, LND and PSM. In terms of operative time, the authors observed that a median OT < 400 min was reached only in the third group, and in the first 33 cases the rate of transfusions and high grade complications (Group 1: 21% vs. Group 3: 3%) was higher when compared to the second and third group. The present study presents major limitations considering that the authors mixed various forms of UD (17% being UCS) and did not use a standardized assessment of the learning curve [23]. 

Guru et al. divided 100 RARC procedures into four groups [24]. Overall OT decreased from 375 min in group 1 to 352 min in group 4, with less than 1% change in OT after case 16. Time from incision to bladder extirpation decreased from 187 min in cohort one to 165 min in cohort 4 [24]. Time for PLND increased from 44 min in cohort 1 to 77 min in cohort 4. LNY increased from 14 nodes in cohort 1 to 23 nodes in cohort 4. Positive surgical margins decreased from four patients in cohort 1 to zero patients in cohort 4. The complication rate had no change from nine patients in cohort 1 to nine patients in cohort 4. Lastly, Dell’Oglio et al. evaluated 164 patients to assess the role of surgical experience on surgical and oncological outcomes. According to their results, surgical experience affects perioperative and oncological outcomes after RARC with iN in a linear fashion, and its beneficial effect does not reach a plateau. Conversely, after 50 cases no further improvement was seen for OT [25]. 

The assessment of the learning curve of RARC with iN is of outmost importance given the complexity of the procedure to ensure patients’ safety and outcomes. When assessing the learning curve of a surgical procedure it is important to select appropriate outcomes. In our study we selected operative time, haemoglobin drop, complications, lymph node count and PSM as main outcomes to assess the learning curve. 

Operative time is often selected as a proxy for learning curve in surgery. Recently, Dell’Oglio et al. suggested that 50 procedures are needed to assess a plateau. In our experience the plateau was reached earlier, at 20 procedures, however we defined the plateau as three consecutive procedures below 300 min. Although operative time is a good proxy for learning curve, it may vary and cannot be considered the only proxy of surgical expertise. 

An important proxy to evaluate during the learning curve is the rate of high-grade complications. In our experience, 33% of the patients presented a complication, however only 12% presented a high-grade complication. We observed a higher rate of complications in the first cases and a plateau is reached after the first 60 cases. Our results are in line with the peer reviewed literature. 

Although the literature suggests that haemoblobin drop, PSM and number of lymph nodes are important factors to evaluate the learning curve of RARCi, according to our results these variables did not change significantly along the learning curve. 

Our results should be interpreted carefully considering previous studies evaluating the outcomes of RARC. One of the largest cohorts evaluating the learning curve of RARC was published by Hayn and coworkers. Overall, the authors evaluated 496 RARC performed by 21 different surgeons reporting a relatively low number of procedures to achieve the learning curve, more specifically 21 cases to reach a plateau for operative time and 30 cases to reach 20 lymph-nodes and to have a 5% overall PSM rate. However, these results are difficult to compare with our study considering that most of these patients underwent the ECUD approach [22]. 

Another study evaluated 67 patients undergoing RARCi observed a beneficial effect of surgical expertise on perioperative outcomes. According to their results, surgical experience improves OT, overall complications and length of stay. However, the results of their study should be interpreted with caution considering the small sample size. Similarly to our results, they found no differences in terms of PSM and LND. 

The present study has several differences with the available literature on the subject. First, we used a standardized method to assess the learning curve which has been widely validated. The CUSUM method has been used in urology and in several other surgical disciplines. Moreover, we focused only on patients undergoing RC with iN and we did not mix different UD, which would have biased our analysis. Lastly, we evaluated an experienced surgeon in laparoscopic and robotic surgery, which is important considering that usually RC are approached after an initial learning curve on robotic kidney and prostatic surgery. 

The robotic approach may have several advantages including reduced hospital stay, blood loss and surgical site infection when compared to the open approach [8]. However, surgeons should keep in mind that the first goal of RC is oncological and according to the available literature, open surgery has similar oncological outcomes when compared to the robotic approach [7,26]. A patient-centered approach should be preferred, and benefits/harms of different surgical techniques and diversions should be thoroughly discussed with the patients before surgery [7]. In the past years the number of robotic platforms around the world has increased dramatically and the number of RARC centers has increased. The key to improve surgical outcomes of RARC lies in the selection of an adequate patient, on the standardization of the surgical technique, and on an adequate postoperative management following the ERAS fast track protocol [20]. In the near future, artificial intelligence, 3D models and virtual reality software may help surgeons when performing RARC, particularly during the learning curve [27]. 

We have to acknowledge some limitations to our study. First, we performed a study including a single surgeon, which may be considered a possible bias. Secondly, the surgeon evaluated presented important surgical expertise in laparoscopic and robotic surgery, which may be considered an advantage. Another possible limitation was the lack of a specific learning curriculum, however to date no validated surgical learning curriculum is available. Another possible limitation is that NIRF imaging and low-pressure pneumoperitoneum were not adopted along these first 100 cases. Lastly, we performed a retrospective analysis, which may be considered a limitation; however, to limit this bias data was collected prospectively. Notwithstanding these limitations, the present study evaluates for the first time the learning curve of RARC with iN using the CUSUM method. 

## 5. Conclusions

Robotic assisted radical cystectomy with intracorporeal urinary diversion is an effective procedure with acceptable morbidity during the learning curve. According to our experience, the learning curve of RARC with iN needs a minimum of 60 procedures to reach the pentafecta. 

## Figures and Tables

**Figure 1 jcm-10-05969-f001:**
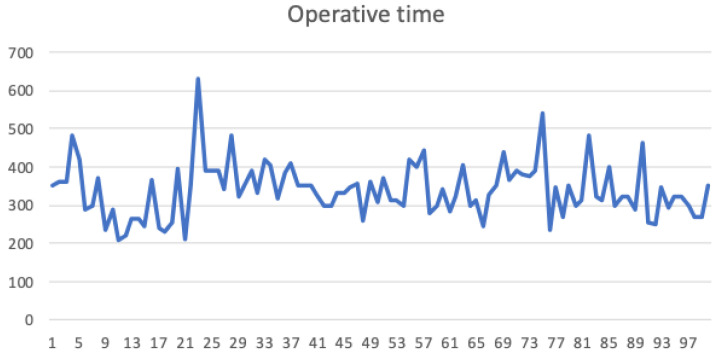
Operative time in consecutive cases.

**Figure 2 jcm-10-05969-f002:**
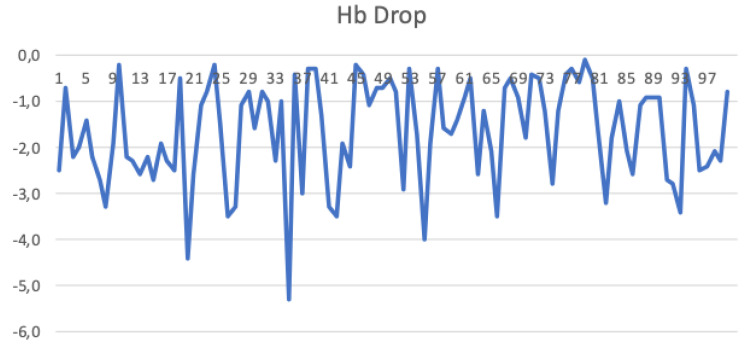
Hb drop in consecutive cases.

**Figure 3 jcm-10-05969-f003:**
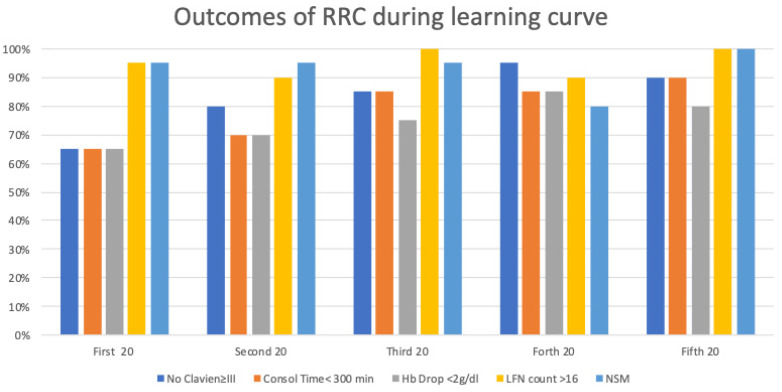
Learning curve according to pentafecta.

**Table 1 jcm-10-05969-t001:** Characteristics of the cohort.

Age (years)	62 (57/66)
Body Mass Index (kg/m^2^)	26 (24/27)
Haemoglobin (Hb) preop (g/dl)	13.6 (11.8/14.5)
Hb Drop (g/dl)	−2.7 (−3.7/−2.0)
Number of lymphnodes	32 (24/38)
Operative Time (min)	330 (300/378)
Lenght of stay (days)	10 (9/14)
pT3 > 3°	28/100 (28%)
pN+	22/100 (22%)
Neoadiuvant Chemotherapy	37/100 (37%)

**Table 2 jcm-10-05969-t002:** Complications according to Clavien classification system.

Complications	26 Patients
Clavien I Mild Hypoxemia Catheter obstruction Other	9/100 3/100 3/100 3/100
Clavien II Fever Anemia requiring trasfusion Acute respiratory failure	23/100 15/100 6/100 2/100
Clavien IIIa Urine leakage requiring nephrostomy placement	7/100
Clavien IIIb Bowel Leakage	9/100
Clavien IV Sepsis and acute kidney failure	1/100
Clavien V	0/100

**Table 3 jcm-10-05969-t003:** Reasons for readmission.

Ureteral stent placement	15/100
Reimplantation	13/100
Nephrostomy placement	4/100
Fever	2/100

## Data Availability

Data is available upon request.

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
