# Peer review of "Benchmarking PASADENA Consensus along the Learning Curve of Robotic Radical Cystectomy with Intracorporeal Neobladder: CUSUM Based Assessment"

_jcm, 2021, doi:10.3390/jcm10245969_

Round 1
Reviewer 1 Report
The article is very interesting: the way of learning to trade is currently changing. There are more and more robots to operate in the world. For this reason, the way in which it is learned or taught is key in the results of the treatments. However, there is an important variable that is not discussed anywhere in the article: the surgeon: age, experience, previous training. The data relating to the surgeons must be included. Otherwise we don't know how you include that variable and it may be a bias. DECISION: ACCEPT WITH MINIMUM CHANGES.
Author Response
We thank the reviewer for his/her comment and for the possibility to improve our manuscript. We agree that the surgeon represents an important variable in the learning curve. In this study we evaluated the learning curve of a single surgeon over 65 years old with an excellent laparoscopic and robotic background. More specifically before starting the learning curve of the robotic cystectomy he had performed >1000 robotic procedures, including radical prostatectomies and partial nephrectomies. Notably, he also performed >1000 open radical cystectomies. As suggested by the reviewer the methods and discussion sections have been improved as follows.
See methods section: ‘All RARC were performed by a single expert open, laparoscopic and robotic surgeon over 65 years old with an extensive previous background including more than 1000 robotic procedures and more than 1000 open cystectomies’.
See limitation section: ‘First we performed a study including a single surgeon which may be considered a possible bias. Secondly, the surgeon evaluated presented an important surgical expertise in laparoscopic and robotic surgery which may be considered an advantage.’
Reviewer 2 Report
In this paper the authors set out to assess the learning curve of RARC-iN and to test its performance in benchmarking Pasadena consensus outcomes. The paper is well written.
I would recommend a revision addressing the following points
- Please highlight the limitation in the discussion section about the data being retrospective.
- In the materials and methods section please could the authors highlight whether any change in technique took place such as the use of ICG in the unit. Were any changes made to the technique including the use of low pressure pneumoperitoneum which has shown significant benefits.
- Please describe the anaesthetic protocol being used during surgery and any enhance recovery specific steps being taken
- Please state how many patients had neoadjuvant chemotherapy / immunotherapy pre RALC
- The bowel leak rate if high at 9%. What technique was used for the bowel anastomosis ? Staple ? Hand sewn
- Please describe the location of the positive surgical margins
- In the discussion section could the authors highlight what steps can be used to improve the outcomes on a neobladder in the peri-operative context for the benefit of those trying to learn and start the technique off.
Congratulations on a great paper.
Author Response
In this paper the authors set out to assess the learning curve of RARC-iN and to test its performance in benchmarking Pasadena consensus outcomes. The paper is well written.
I would recommend a revision addressing the following points
- Please highlight the limitation in the discussion section about the data being retrospective.
We thank the reviewer for his/her comment. The limitation section has been improved as follows.
‘Lastly, we performed a retrospective analysis which may be considered a limitation however to limit this bias data was collected prospectively.’
- In the materials and methods section please could the authors highlight whether any change in technique took place such as the use of ICG in the unit. Were any changes made to the technique including the use of low-pressure pneumoperitoneum which has shown significant benefits.
We thank the reviewer for his/her comment. The technique was not modified along the learning curve and unfortunately in this first 100 cases NIRF imaging and low pneumo-pressure were not adopted. We agree that both techniques have shown significant benefits and therefore the methods and limitations section has been improved.
See Methods section: ‘The technique was not modified along the learning curve, NIRF imaging and low-pressure pneumoperitoneum were not used’.
See Limitations section: Another possible limitation is that NIRF imaging and low-pressure pneumoperitoneum were not adopted along these first 100 cases.
- Please describe the anaesthetic protocol being used during surgery and any enhance recovery specific steps being taken
We thank the reviewer for the possibility to improve our manuscript, patients underwent surgery under general anesthesia and followed the fast track/ERAS protocol as described by Karl et al in 2014. Manuscript has been improved as follows in the methods section.
Methods section: The interventions were performed under general anesthesia and the fast-track ERAS protocol was applied as described by Karl et al[20].
- Please state how many patients had neoadjuvant chemotherapy / immunotherapy pre RALC
We thank the reviewer for his/her comment and for the possibility to improve our manuscript. Overall 37/100 patients underwent neoadiuvant chemotherapy while none underwent immunotherapy. Manuscript results section has been updated.
- The bowel leak rate if high at 9%. What technique was used for the bowel anastomosis ? Staple ? Hand sewn
We thank the reviewer for his/her comment. Bowel anastomosis was performed using staplers. This is now specified in the methods section.
See methods section: The division of the proximal ileum was made using only one stapler load (60 mm). Isolation of the distal extremity of the ileal segment was carried out with a 6- to 8-cm deep section of the mesentery using two consecutive stapler loads (60 mm and 45 mm).
- Please describe the location of the positive surgical margins
We thank the reviewer for his/her comment. All positive surgical margins were found on the ureters which were sent for frozen section while no soft tissue positive surgical margins were recorded.
- In the discussion section could the authors highlight what steps can be used to improve the outcomes on a neobladder in the peri-operative context for the benefit of those trying to learn and start the technique off.
We thank the reviewer for his/her comment. Discussion section has been improved as follows.
See Discussion section:
The robotic approach may have several advantages including reduced hospital stay, blood loss and surgical site infection when compared to the open approach. However, surgeons should keep in mind that the first goal of RC is oncological and according to the available literature open surgery has similar oncological outcomes when compared to the robotic approach. A patient centered approach should be preferred, and benefits/harms of different surgical techniques and diversions should be deeply discussed with the patients before surgery. In the past years the number of robotic platforms around the world has increased dramatically as well as high volume RARC centers. The key to improve surgical outcomes of RARC lies on the selection of the adequate patient, on the standardization of the surgical technique and on an adequate postoperative management following the ERAS fast track protocol. In the near future, artificial intelligence, 3d models and virtual reality software’s may probably help surgeons when performing RARC specially during the learning curve.
Reviewer 3 Report
In their study, the authors benchmark PASADENA consensus along the learning curve of robotic radical cystectomy (RARC) with intracorporeal orthotopic diversion. This surgical procedure is a challenging one and high expertise in the field is requested.
The choice of the surgical technique shall take into account a complex range of factors and decision-making by both physicians and patients (doi:10.1186/s12885-020-07748-7). Robotic cystectomy could reduce the hospital of stay, blood loss as well surgical site infection (doi: 10.1515/med-2019-0081). As well for other cancers, oncological results are mandatory before any others (doi: 10.21037/jtd.2018.07.21). Open radical cystectomy remains the most diffused technique but the robotic approach is gaining every day more space thanks also the high volume center with expert robotic surgeons (doi: 10.1097/MOU.0000000000000930).
The manuscript is well developed but should be improved before a publication in the fields suggested.
Author Response
In their study, the authors benchmark PASADENA consensus along the learning curve of robotic radical cystectomy (RARC) with intracorporeal orthotopic diversion. This surgical procedure is a challenging one and high expertise in the field is requested.
The choice of the surgical technique shall take into account a complex range of factors and decision-making by both physicians and patients (doi:10.1186/s12885-020-07748-7). Robotic cystectomy could reduce the hospital of stay, blood loss as well surgical site infection (doi: 10.1515/med-2019-0081). As well for other cancers, oncological results are mandatory before any others (doi: 10.21037/jtd.2018.07.21). Open radical cystectomy remains the most diffused technique but the robotic approach is gaining every day more space thanks also the high volume center with expert robotic surgeons (doi: 10.1097/MOU.0000000000000930).
The manuscript is well developed but should be improved before a publication in the fields suggested.
We thank the reviewer for his/her suggestion and for the possibility to improve our manuscript. Discussion section has been improved taking into account references deserving discussion.
For instance, the reference
- doi: 10.21037/jtd.2018.07.21 is a report on surgical outcomes of lung adenocarcinoma. There is no reason at all for considering such reference as a way to improve discussion. Therefore, this point was discussed with the proper reference (Perioperative and mid-term oncologic outcomes of robotic assisted radical cystectomy with totally intracorporeal neobladder: Results of a propensity score matched comparison with open cohort from a single-centre series. Simone G, Tuderti G, Misuraca L et al. doi.org/10.1016/j.ejso.2018.04.006).
- The reference used to improve discussion about the need for considering both physicians and patients perspectives was not the doi:10.1186/s12885-020-07748-7 but the DOI: 10.1016/j.euf.2021.03.002, which is a level 1 evidence coming from a RCT comparing open vs robotic cystectomies with intracorporeal urinary diversions.
The reference doi: 10.1097/MOU.0000000000000930 is a commentary reporting “as the learning curve for ICUD diversion has flattened, retrospective analyses have emerged that suggest this technique may hold benefit over both ORC and RARC with ECUD, though current data is conflicting, and a randomized controlled study is forthcoming.” Therefore, ongoing randomized controlled trials were reported to highlight the need for evidences from RCTs (Mastroianni et al Eur Urol Focus, NCT NCT03434132, Catto et al NCT03049410)
- Eventually, suggesting discussion about the reference doi: 10.1515/med-2019-0081 is totally misleading, being a short report on 60 patients who underwent “minimally invasive surgery” in a “so called” high volume center reporting outcomes on 13 radical cystectomies performed in a 6mo time frame, two robotic and 11 open surgeries. Reviewers should omit any suggestion about self-citation if not focused on the topic and moreover if quality of papers suggested is far from being considered high level evidences.
We reject any of these suggestions and welcome a reply from the reviewer.
See Discussion section:
The robotic approach may have several advantages including reduced hospital stay, blood loss and surgical site infection when compared to the open approach[25]. However, surgeons should keep in mind that the first goal of RC is oncological and according to the available literature open surgery has similar oncological outcomes when compared to the robotic approach[7,26]. A patient centered approach should be preferred, and benefits/harms of different surgical techniques and diversions should be deeply discussed with the patients before surgery[7]. In the past years the number of robotic platforms around the world has increased dramatically as well as high volume RARC centers. The key to improve surgical outcomes of RARC lies on the selection of the adequate patient, on the standardization of the surgical technique and on an adequate postoperative management following the ERAS fast track protocol[20]. In the near future, artificial intelligence, 3d models and virtual reality software’s may probably help surgeons when performing RARC specially during the learning curve[27].